Colvin et al. Implementation Science 2018, **13**(Suppl 1):13

Implementation Science

# Applying GRADE-CERQual to qualitative evidence synthesis findings—paper 4: how to assess coherence

Christopher J. Colvin[1], Ruth Garside[2], Megan Wainwright[1], Heather Munthe-Kaas[3*], Claire Glenton[3], Meghan A. Bohren[4], Benedicte Carlsen[5], Özge Tunçalp[4], Jane Noyes[6], Andrew Booth[7], Arash Rashidian[8,9], Signe Flottorp[3] and Simon Lewin[3,10]

## Abstract

**Background:** The GRADE-CERQual (Grading of Recommendations Assessment, Development and Evaluation-Confidence in Evidence from Reviews of Qualitative research) approach has been developed by the GRADE working group. The approach has been developed to support the use of findings from qualitative evidence syntheses in decision-making, including guideline development and policy formulation.
CERQual includes four components for assessing how much confidence to place in findings from reviews of qualitative research (also referred to as qualitative evidence syntheses): (1) methodological limitations, (2) relevance, (3) coherence and (4) adequacy of data. This paper is part of a series providing guidance on how to apply CERQual and focuses on CERQual's coherence component.

**Methods:** We developed the coherence component by searching the literature for definitions, gathering feedback from relevant research communities and developing consensus through project group meetings. We tested the CERQual coherence component within several qualitative evidence syntheses before agreeing on the current definition and principles for application.

**Results:** When applying CERQual, we define coherence as how clear and cogent the fit is between the data from the primary studies and a review finding that synthesises that data. In this paper, we describe the coherence component and its rationale and offer guidance on how to assess coherence in the context of a review finding as part of the CERQual approach. This guidance outlines the information required to assess coherence, the steps that need to be taken to assess coherence and examples of coherence assessments.

**Conclusions:** This paper provides guidance for review authors and others on undertaking an assessment of coherence in the context of the CERQual approach. We suggest that threats to coherence may arise when the data supporting a review finding are contradictory, ambiguous or incomplete or where competing theories exist that could be used to synthesise the data. We expect the CERQual approach, and its individual components, to develop further as our experiences with the practical implementation of the approach increase.

**Keywords:** Qualitative research, Qualitative evidence synthesis, Systematic review methodology, Research design, Methodology, Confidence, Guidance, Evidence-based practice, Coherence, GRADE

\* Correspondence: heather.munthe-kaas@fhi.no
[3]Norwegian Institute of Public Health, Oslo, Norway
Full list of author information is available at the end of the article

## Background

The GRADE-CERQual (Confidence in Evidence from Reviews of Qualitative research) approach has been developed by the GRADE (Grading of Recommendations Assessment, Development and Evaluation) working group. The approach has been developed to support the use of findings from qualitative evidence syntheses in decision-making, including guideline development and policy formulation.

GRADE-CERQual (hereafter referred to as CERQual) includes four components for assessing how much confidence to place in findings from reviews of qualitative research (also referred to as qualitative evidence syntheses): (1) methodological limitations; (2) relevance; (3) coherence; and (4) adequacy of data. This paper focuses on one of these four CERQual components: coherence.

When carrying out a CERQual assessment, we define the coherence of the review finding as how clear and cogent the fit is between the data from the primary studies and a review finding that synthesises that data. By "cogent" we mean well supported or compelling. For more descriptive review findings, a 'coherent' finding would represent well the underlying patterns that appear in the data. For more interpretive or explanatory review findings, a 'coherent' finding would provide a strong account of the patterns in the data through convincing interpretations or explanations. Later in this paper, we describe in more detail how we conceptualise the spectrum of more descriptive to more explanatory findings. When

the fit between the data from primary studies and the review finding that synthesises that data is not fully clear and cogent, we are less confident that the finding reflects the phenomenon of interest. The coherence component in CERQual is analogous to the inconsistency domain used in the GRADE approach for findings from systematic reviews of effectiveness [1].

## Aim

The aims of this paper, part of a series (Fig. 1), are to describe what we mean by the coherence of a review finding in the context of a qualitative evidence synthesis and to give guidance on how to operationalize this component in the context of a review finding, as part of the CERQual approach. This paper should be read in conjunction with the papers describing the other three CERQual components [2–4] and the paper describing how to make an overall CERQual assessment of confidence and create a Summary of Qualitative Findings table [5]. Key definitions for the series are provided in Additional file 1.

## How CERQual was developed

The initial stages of the process for developing CERQual, which started in 2010, are outlined elsewhere [6]. Since then, we have further refined the current definitions of each component and the principles for application of the overall approach using a number of methods. When developing CERQual's coherence component, we undertook

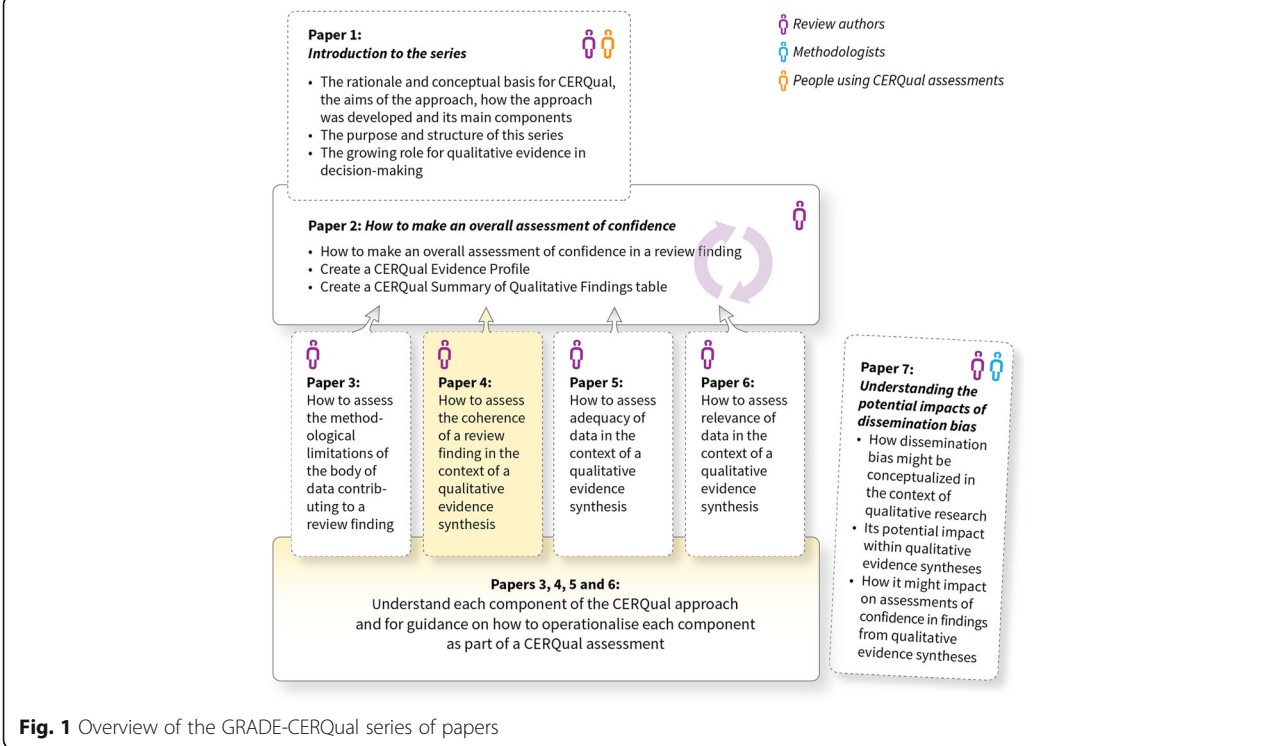

**Fig. 1** Overview of the GRADE-CERQual series of papers

informal searches of the literature, including Google and Google Scholar, for definitions and discussion papers related to the concept of coherence and to related concepts such as transformation of findings, descriptive findings and explanatory findings. We carried out similar searches for the other three components. We presented an early version of the CERQual approach in 2015 to a group of methodologists, researchers and end users with experience in qualitative research, GRADE or guideline development. We further refined the approach through training workshops, seminars and presentations during which we actively sought, collated and shared feedback; by facilitating discussions of individual CERQual components within relevant organisations; through applying the approach within diverse qualitative evidence syntheses [7–17]; and through supporting other teams in using CERQual [18, 19]. As far as possible, we used a consensus approach in these processes. We also gathered feedback from CERQual users an online feedback form and through short individual discussions. The methods used to develop CERQual are described in more detail in the first paper in this series [20].

## Assessing coherence

The coherence of a review finding is an assessment of how clear and cogent the fit is between the data from the primary studies and a review finding that synthesises that data. In both primary qualitative research and qualitative evidence syntheses, 'findings' are 'transformations' of the underlying data into descriptions, interpretations and/or explanations of the phenomenon of interest. Qualitative evidence synthesis findings are developed by identifying patterns in the data across the primary studies included in the synthesis.

In qualitative evidence syntheses, as in primary qualitative research, one can think of findings as being located along a continuous spectrum representing the degree of transformation of the data [21] (Fig. 2, adapted from [21]). At one end of the spectrum are more descriptive findings, i.e. findings that describe patterns in the data. At the other end of the spectrum are interpretive or explanatory findings. These transformed findings provide theoretical interpretations or explanations of the patterns in the data (for examples, see Table 1). Between these two poles are findings that do more than simply describe the data but are not yet themselves full-fledged

interpretations or explanations. These findings may explore patterns of association in the data and/or link patterns in the data to key theoretical concepts. The terms above the line in Fig. 2—thematic survey, conceptual/thematic description and interpretive explanation—are the terms used by the original authors to illustrate the different kinds of findings along this spectrum of data transformation.

When assessing coherence, it is important to consider the difference between more descriptive review findings and more explanatory review findings. While some qualitative evidence synthesis methods tend to produce more findings at one end of the spectrum than the other (e.g. meta-aggregation, which produces more descriptive review findings, and meta-ethnography, which produces more explanatory review findings), it is often the case that a qualitative evidence synthesis will include a mix of more descriptive and more explanatory findings. Wherever a review finding falls on the spectrum, however, a CERQual assessment of coherence asks the same broad question—is the fit between the underlying data from the primary studies and the review finding clear and cogent?

The ways in which this fit is assessed will vary by the type of review finding being assessed. Descriptive findings provide a summary of the underlying patterns of data in the studies. When these underlying patterns are complex or varied, the coherence of a descriptive review finding depends on how clearly and cogently this complexity and variation is described in the review finding. The coherence of a descriptive finding may be threatened, however, if it only describes the most dominant patterns in the data and does not sufficiently capture the presence of 'outliers' and/or ambiguous elements in the data. By outlier, we refer to data in underlying studies that do not fit the dominant data patterns across the studies.

More explanatory review findings offer interpretations or explanations of patterns in the data. The coherence of an interpretive or explanatory finding depends on how clearly and cogently these patterns are interpreted or explained in the finding. The coherence of this kind of finding may be threatened by the presence of data in the primary studies that challenge the main interpretation or explanation in the review finding ('disconfirming cases') or by plausible competing interpretations or explanations.

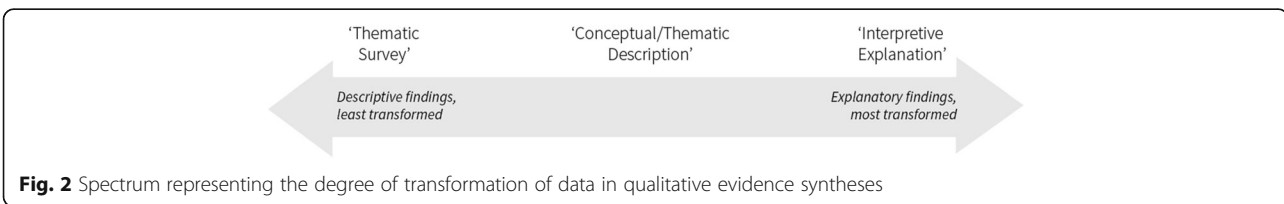

**Fig. 2** Spectrum representing the degree of transformation of data in qualitative evidence syntheses

**Table 1** CERQual assessments of coherence for different kinds of review findings—examples

| Review findings | Concerns about coherence |
| --- | --- |
| Descriptive review findings | |
| Women are comfortable with the process of managing medical abortion at home | *Moderate concerns:* though generally the case, the data were actually more varied and this finding is an over-simplified description of the underlying patterns of comfort/discomfort. |
| The experience of women having a medical abortion at home varied. Some felt overwhelmed, some felt comfortable and empowered, and some reported that it was just like any other minor medical procedure | *Minor concerns:* the data were indeed varied, and these were three broad types of discomfort expressed by women. The studies usually addressed this issue in passing, though, and did not often explore in detail what women meant when they said they expressed comfort, empowerment or feeling overwhelmed. |
| Conceptual review findings | |
| Most women who were counselled by trained medical providers had a good experience with medical abortion. When women who had been counselled by trained professionals had a bad experience, it was because of 'disrupted expectations', when the experience did not match what they were told to expect. | *No or very minor concerns:* the finding reflects the complexity and variation of the data, and the association of bad experiences with 'disrupted expectations' is well supported by details in the underlying studies. We explored other possible explanations for bad experiences despite the provision of counselling (e.g. poor or inconsistent counselling by trained medical providers) but found no data supporting these alternatives. |
| Interpretive/explanatory review findings | |
| When women have a sense of self-efficacy and control, have access to information and emergency health services, trust their providers and have appropriately trained providers, their experience of medical abortion at home is positive. The sense of self-efficacy and control and their trust in providers are the most important factors in their experience but these cannot be introduced at the time of the abortion services (i.e. they have to already be in place) | *Serious concerns:* the interpretation in this finding is somewhat supported by data from several studies. However, there were some contradictory cases that did not fit the model in the finding (e.g. one study where women met the model's criteria but nonetheless reported a poor experience of medical abortion at home). In other studies, it was hard to tell if the data really supported this model because of vaguely defined measures or inconsistent definitions across studies. |

Assessing the fit between the data and a review finding will therefore necessarily involve review authors looking actively for data that complicate or challenge their main review findings [22]. This iterative analytic approach is typical in qualitative evidence syntheses. In the process, review authors may identify problems with how the review finding itself was formulated and may make modifications to their review findings to strengthen the fit between review finding and data. This is also a chance to check whether initial review findings have inappropriately oversimplified (or "smoothed out") data or stretched an explanation too far.

Note, however, that when assessing coherence for CERQual, our aim is not to judge whether some absolute standard of coherence has been achieved, but to judge whether there are grounds for concern regarding coherence that are serious enough to lower our confidence in the review finding.

**Balancing the coherence and utility of review findings**

Since the review authors identify and organise the patterns that constitute a review finding, assessing coherence during the synthesis offers an opportunity for reflection on that process. By being guided to specifically examine the coherence of each review finding, the review authors are given the opportunity to reflect critically on the extent to which the pattern (review finding) really represents a strong fit with the underlying data.

In this process, however, review authors might revise a review finding in ways that strengthen its coherence but

limit its usefulness for users of the review. Review authors could, for example, strengthen the coherence of a review finding by reframing it in more general, vague or equivocal terms, or alternatively, in a highly specified fashion such that it only applies to a very limited number of cases. These kinds of descriptive findings may be coherent (i.e. strongly supported by the data) but may be of limited utility since they have been either too broadly, vaguely or narrowly framed.

Review authors might also strengthen the coherence of their review finding by avoiding more transformed interpretive or explanatory findings in favour of more descriptive findings that have fewer threats to their coherence. Again, these kinds of descriptive findings may be coherent but may be of less utility since they fail to offer users of the review any explanations for the patterns described.

There are circumstances where it is important to report a review finding from a qualitative evidence synthesis because of its potential utility to readers, even though there are serious concerns about the coherence of that review finding. Some example situations in which findings with concerns about coherence may nonetheless be useful include:

- Highlighting less frequent, or poorly understood but nonetheless potentially important phenomena
- Highlighting novel or surprising review findings that challenge conventional perspectives
- Ensuring that under-researched or marginalised populations, settings or experiences are not disregarded

- Developing more integrative and theoretical accounts that can help policy-makers and programme managers consider the role of local phenomenon, relationships, processes and contexts
- Answering an explicit, pre-defined question of interest to review authors, policy-makers or practitioners

While the process of assessing the coherence of a review finding during a qualitative evidence synthesis can encourage critical reflection and refinement of review findings, review authors should also ensure that review findings do not prioritise coherence at the expense of utility. Where possible, review authors should aim to maximise both.

### Guidance on how to assess coherence in the context of a review finding

The steps taken when assessing coherence are shown in Fig. 3 and detailed below. As described above, these steps may be iterative, particularly if there are serious concerns about the coherence of preliminary findings, to ensure, for example, that nuances in the data are appropriately captured in the findings.

### Step 1: collect and consider the necessary information related to coherence

To assess the coherence of a review finding, you will need access to the underlying data contributing to the review finding. This will normally be available in the data extraction tables produced as part of the review process. The assumption is that all data relevant to the review finding—including data that did not fully support the review finding but were relevant to the topic of the review finding—were extracted. If not, then it may be

**Fig. 3** Steps in assessing the coherence of a review finding

necessary to return to the primary studies themselves when assessing coherence. It may also be necessary to return to the primary studies, or develop further coding, if details necessary for assessing how well the data support a particular review finding were not originally captured in the extraction tables.

For more interpretive or explanatory review findings, you may also need information on the concepts and theories used to develop, or developed from, the review finding. Theories used in qualitative evidence syntheses may include:

a. Theories imported from the existing literature, external to the papers included in the synthesis
b. Theories developed from the theory used in one (or more) of the papers included in the synthesis and then applied across findings from other papers
c. Theories developed as an original explanation or interpretation by the review authors during the synthesis process.

In many cases, a qualitative evidence synthesis may include review findings using theory from all three of these sources. When theory is used in review findings to explain underlying patterns in the data, review authors should specify whether the theory is imported, identified in the included studies or original. Those using CERQual to assess the coherence of these review findings will need sufficient information about these theories in order to assess how clearly and cogently they explain the underlying data.

If you are using CERQual on findings from your own review, you should already have easy access to all of this information. However, if you are assessing the coherence of findings from other people's reviews, collecting this information is likely to be a time-consuming process. At present, review authors do not commonly report all of the data that has led to each review finding. Unless you have access to their data extraction sheets or coding files, you will need to trace this data by following the references associated with each review finding. For more information on applying CERQual to findings from somebody else's review, see paper 2 in this series [5].

### Step 2: assess the body of data that contributes to each finding and decide whether you have concerns about coherence

Once you have collected the information you need, you can start to assess if there are any threats to how clear and cogent the fit is between each review finding and the data related to that review finding. When there is clear and cogent support for a review finding across the underlying data, you should not have serious concerns about the coherence of the finding. You may have a

concern about coherence of the fit between a review finding and the underlying data when patterns in the data are not well explored or explained, either by the review authors or by the primary study authors.

We have identified three types of threats to coherence—contradictory data, ambiguous or incomplete data and competing theories. You should identify threats to coherence when:

a) Some of the data from included studies *contradict* the review finding. For example,

- In a review finding that is primarily descriptive, some elements of the data from included studies might not fit the description of the key patterns captured in the finding. These contradictory data—what might be termed 'outliers'—may have been omitted in the review finding because review authors either wanted to highlight only the dominant patterns or were addressing a specific policy or guideline question that required a more narrow response. In these cases, the evidence that is not well captured within the review finding may be considered a threat to coherence. CERQual users will have to judge how serious a concern they consider these outliers to be.
- In a review finding that is more explanatory or interpretive, some elements of the underlying data might conflict with the interpretation or explanation offered in the finding. These data might be thought of as 'disconfirming' or 'contradictory' data. When a review finding can offer a cogent explanation for these conflicting data, we would not consider this a threat to coherence.

b) It is *not clear* if some of the underlying data support the review finding. For example,

- Key aspects of the underlying data may be vaguely defined or described. In these cases, the supporting data are not clearly or sufficiently described and we cannot always be sure that the data in fact clearly support the review finding
- Elements of the underlying data may be defined in slightly different ways across different studies. In these cases, the data may appear reasonably comparable but we are not sure if they are in fact comparable
- More interpretive or explanatory review findings are often more complex and include a number of aspects, e.g. descriptive data, ideas, concepts or relationships. We may have strong evidence from the underlying data for certain aspects of the review finding, but insufficient data to support other

aspects of the interpretation or explanation. These gaps in the evidence for an interpretive or explanatory review finding are not contradictory data, but rather the absence of data in certain places. When the data provide this kind of incomplete support for a review finding, you may have concerns about the coherence of a finding. Gaps may be less important when researchers are "importing" a theory from the existing literature that is already very well established and developed. For example, if the concept of stigma is used to explain why some people hide their mental health status, this is such a well-developed social theory that the coherence of this as an explanatory review finding may not be threatened, even if not all aspects of stigma are identified in the evidence synthesised

c) *Plausible alternative* descriptions, interpretations or explanations could be used to synthesise the underlying data. In these cases, the concern is not that there is not a clear fit between data and review finding per se. Rather, the concern is that there are other alternative plausible ways of describing, interpreting or explaining the data, and these competing theories have not been explored or assessed by the review authors.

**Step 3: make a judgement about the seriousness of your concerns and justify this judgement**

Once you have assessed coherence for each review finding, decide whether any concerns that you have identified should be categorised as either:

- No or very minor concerns
- Minor concerns
- Moderate concerns
- Serious concerns

You should begin with the assumption that there are no concerns regarding coherence. In practice, minor concerns will probably not lower our confidence in the review finding, while serious concerns are likely to lower our confidence. Moderate concerns may lead us to consider lowering our confidence in a review finding where there are also concerns in relation to other CERQual components.

Where you have concerns about coherence, you should describe these concerns in the CERQual Evidence Profile in sufficient detail to allow users of the review findings to understand the reasons for the assessments made. The Evidence Profile presents each review finding along with the assessments for each CERQual component, the overall CERQual assessment for that finding and an explanation of this overall assessment. For more information, see the second paper in this series [5].

Review authors may also want to note the extent to which they have explored other plausible alternative explanations. Your assessment of coherence will be integrated into your overall assessment of confidence in each review finding. How to make this overall assessment of confidence is described in the second paper in this series [5].

### Examples of assessing coherence

In Table 1, we give some examples of how coherence can be assessed for a selection of review findings. These examples illustrate how assessments of coherence can operate across the spectrum of types of findings described above. The examples are adapted from a recent qualitative evidence synthesis on medical abortion and efforts to 'task shift' elements of the medical abortion process from the clinical space to the home context where possible [23, 24].

The first two review findings are based on the same data and show how it is possible to construct different findings that, in turn, are subject to different types of threats and may raise varying degrees of concern about coherence. The first is an overly simplified representation of the data in the studies, and the second is a more nuanced formulation of the data that is based on ideas that often went under-explored in the primary studies.

Note that within the context of a review that sets out to aggregate information in a synthesis, the first two descriptive review findings may be reasonable outputs. However, these descriptive findings may not be the most useful for policy-makers and practitioners. While they describe the range of experiences, no attempt is made to explain them or to interpret implications of such variation.

### Implications when concerns regarding coherence are identified

Concerns about coherence may not only have implications for our confidence in a review finding, but can also point to ways of improving future research. Firstly, these concerns may suggest that more primary research needs to be done in that area. This additional research may require more data and/or more analysis/interpretation of existing data. The review team should also consider whether the review needs updating once this research is available.

Secondly, review authors should consider using the patterns found across primary studies to generate new hypotheses or theory regarding the issue addressed by the finding. For example, the hypotheses in the last review finding in Table 1 about the key factors affecting women's comfort with medical abortion at home may provide a direction for future research.

Finally, when a review has not included all potential studies but has instead used a sampling procedure to select studies for inclusion, future updates of the review could reconfigure the sampling to explore the variation found. Any changes that are made to the scope of the review are also likely to have an impact on our assessment of the other CERQual components.

### Conclusions

Concerns about coherence may lower our confidence in review findings and are therefore part of the CERQual approach. However, it is also important to remember that coherence is just one component of the CERQual approach. Having concerns about coherence may not necessarily lead to a downgrading of overall confidence in a review finding, as these concerns will be assessed alongside those for the other three CERQual components.

In this paper, we have described our thinking so far and provided guidance to review authors and others on how to assess threats to the coherence of findings from qualitative evidence syntheses. We suggest that concerns to coherence may arise when the data supporting a review finding is contradictory, ambiguous or incomplete or where competing theories that could be used to synthesise the data are left unexplored. We expect the CERQual approach, and its individual components, to develop further as our experiences with the practical implementation of the approach increase.

### Open peer review

Peer review reports for this article are available in Additional file 2.

### Additional files

**Additional file 1:** Key definitions relevant to CERQual. (PDF 619 kb)

**Additional file 2:** Open peer review reports. (PDF 98 kb)

**Acknowledgements**
Our thanks for their feedback to those who participated in the GRADE-CERQual Project Group meetings in January 2014 or June 2015 or gave comments to the paper: Elie Akl, Heather Ames, Zhenggang Bai, Rigmor Berg, Jackie Chandler, Karen Daniels, Hans de Beer, Kenny Finlayson, Signe Flottorp, Bela Ganatra, Stephen Gentles, Susan Munabi-Babigumira, Andy Oxman, Tomas Pantoja, Vicky Pileggi, Kent Ranson, Rebecca Rees, Anna Selva, Holger Schünemann, Elham Shakibazadeh, Birte Snilstveit, James Thomas, Hilary Thompson, Judith Thornton, Joe Tucker and Josh Vogel. Thanks also to Sarah Rosenbaum for developing the figures used in this series of papers and to members of the GRADE working group for their input. The guidance in this paper has been developed in collaboration and agreement with the GRADE working group (www.gradeworkinggroup.org).

**Funding**
This work, including the publication charge for this article, was supported by funding from the Alliance for Health Policy and Systems Research, WHO (http://www.who.int/alliance-hpsr/en/). Additional funding was provided by the Department of Reproductive Health and Research, WHO (www.who.int/reproductivehealth/about_us/en/); Norad (Norwegian Agency for Development Cooperation: www.norad.no), the Research Council of Norway

(www.forskningsradet.no); and the Cochrane methods Innovation Fund. SL is supported by funding from the South African Medical Research Council (www.mrc.ac.za). The funders had no role in study design, data collection and analysis, preparation of the manuscript or the decision to publish.

### Availability of data and materials
Additional materials are available on the GRADE-CERQual website (www.cerqual.org)
To join the CERQual project group and our mailing list, please visit our website: http://www.cerqual.org/contact/. Developments in CERQual are also made available via our Twitter feed: @CERQualNet.

### About this supplement
This article has been published as part of *Implementation Science* Volume 13 Supplement 1, 2018: Applying GRADE-CERQual to Qualitative Evidence Synthesis Findings. The full contents of the supplement are available online at https://implementationscience.biomedcentral.com/articles/supplements/volume-13-supplement-1.

### Authors' contributions
All authors participated in the conceptual design of the CERQual approach. CC, RG and MW wrote the first draft of the manuscript. All authors contributed to the writing of the manuscript. All authors have read and approved the manuscript.

### Ethics approval and consent to participate
Not applicable. This study did not undertake any formal data collection involving humans or animals.

### Consent for publication
Not applicable

### Competing interests
The authors declare that they have no competing interests.

## 
### Author details
[1]Division of Social and Behavioural Sciences, School of Public Health and Family Medicine, University of Cape Town, Cape Town, South Africa. [2]European Centre for Environment and Human Health, University of Exeter Medical School, Exeter, UK. [3]Norwegian Institute of Public Health, Oslo, Norway. [4]UNDP/UNFPA/UNICEF/WHO/World Bank Special Programme of Research, Development and Research Training in Human Reproduction, Department of Reproductive Health and Research, WHO, Geneva, Switzerland. [5]Uni Research Rokkan Centre, Bergen, Norway. [6]School of Social Sciences, Bangor University, Bangor, UK. [7]School of Health and Related Research (ScHARR), University of Sheffield, Sheffield, UK. [8]Department of Health Management and Economics, School of Public Health, Tehran University of Medical Sciences, Tehran, Iran. [9]Information, Evidence and Research Department, Eastern Mediterranean Regional Office, World Health Organization, Cairo, Egypt. [10]Health Systems Research Unit, South African Medical Research Council, Cape Town, South Africa.

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
