## [Open peer review reports. (PDF 98 kb) · Implementation Science : IS]

Open Peer Review reports

Original Submission		
1 Nov 2016	Submitted	Original Manuscript
2 Jan 2017	Reviewed	Reviewer Report - Soo Downe This is a generally well written paper in an area of methodological development that is gaining in importance at all levels of implementation of change. The paper offers clear guidance to the current state of the art in assessing coherence in reviews of qualitative evidence. The only issue that might need attention is the lack of consistency in the use of the first, second, and third person throughout the text. Where the first and second person is used, it seems rather like the paper has been directly adapted from training materials, rather than having been written for an academic journal. This may be the style adopted for the CerQual series, but it seems rather awkward when read as a stand alone paper. Otherwise, I only noticed one grammatical error: p 13: 'while serious concerns are likely lower our confidence.' This paper will be very useful for all those who are working in the field of reviews of qualitative evidence.
20 Jan 2017	Reviewed	Reviewer Report - Elizabeth Nye 'Description of and rationale for the coherence component' * The explanation of the continuum of descriptive to explanatory findings is laid out well for the reader.* The terms 'findings' and 'data' are used regularly to refer to the different levels of qualitative evidence synthesis ('findings') and qualitative primary studies ('data'). Yet, it might not be clear to all readers what exactly is meant by 'data'. Do you refer to the individual accounts from the primary studies (such as illustrated through direct quotations in primary studies), the findings of primary studies (i.e., the descriptions or interpretations reported by the researchers), or could you be referring to both? A brief explanation would be helpful for this section.* Very helpful clarification of what you mean by 'outlier' data and 'disconfirming cases'. 'Balancing the coherence and utility of review findings' * Two examples are provided of ways authors might strengthen the coherence and limit the utility of their review findings: 1) reframing findings vaguely/narrowly, and 2) favouring descriptive over interpretive/explanatory findings. These are both explained nicely.* It would be helpful if, in the paragraph immediately following with a bulleted list of situations in which coherence might be found to be low, that the word 'utility' is explicitly used in the explanation of why these are circumstances 'important to review'. The consistent use of this terminology helps guide readers, particularly those who might be new to the topic. 'Guidance on how to assess coherence in the context of a review finding' * The text is clear in that the steps for assessing coherence might be iterative, however this is not reflected in

the accompanying visual (Figure 3). The figure should be revised to reflect the potential for an iterative process rather than reflecting what appears to be a simple, linear process. Perhaps the addition of dashed arrows looping back to previous steps or something similarly minimalistic would be appropriate to accommodate the style of the current figure.

* Step 1: particularly given the direction of this paragraph describing accessing 'underlying data', the earlier comment addressing a clearer explanation of what is meant by the 'data' is needed at the beginning of the paper.

* Very nice explanation of the three types of theories potentially present in qualitative evidence syntheses.

* Strong section on three types of threats to coherence - the concrete example about stigma and hiding mental health status serves as a really useful tool to illustrate for readers a situation when gaps are not as concerning when researchers import a theory.

* Table 1 and its examples/explanations are very useful and an excellent addition to this paper.

29 Jun 2017

Author responded

Author comments - Heather Munthe-Kaas

Peer reviewer comments	Responses
Reviewer #1:	
This is a generally well written paper in an area of methodological development that is gaining in importance at all levels of implementation of change. The paper offers clear guidance to the current state of the art in assessing coherence in reviews of qualitative evidence. The only issue that might need attention is the lack of consistency in the use of the first, second, and third person throughout the text. Where the first and second person is used, it seems rather like the paper has been directly adapted from training materials, rather than having been written for an academic journal. This may be the style adopted for the CerQual series, but it seems rather awkward when read as a stand alone paper.	Thank you We have checked the paper for consistency in the use of the first, second and third person and tried to make it clear throughout that when we say 'you', we are referring to those applying the CERQual approach. We have used this 'second person' approach where we provide guidance on how to apply CERQual – one of the aims of this series of papers – in order to indicate the more 'instructional' nature of these parts of the paper. In contrast, we have used 'we' when describing what we did to develop each component and how we define each component. In other parts of the paper, we have used the third person.
Otherwise, I only noticed one grammatical error: p 13: 'while serious concerns are likely lower our confidence.'	This has been corrected.
This paper will be very useful for all those who are working in the field of reviews of qualitative evidence.	Thank you
Reviewer #2:	
'Description of and rationale for the coherence component':	Thank you

* The explanation of the continuum of descriptive to explanatory findings is laid out well for the reader.	
* The terms 'findings' and 'data' are used regularly to refer to the different levels of qualitative evidence synthesis ('findings') and qualitative primary studies ('data'). Yet, it might not be clear to all readers what exactly is meant by 'data'. Do you refer to the individual accounts from the primary studies (such as illustrated through direct quotations in primary studies), the findings of primary studies (i.e., the descriptions or interpretations reported by the researchers), or could you be referring to both? A brief explanation would be helpful for this section.	Thank you for this helpful comment. We have now defined both 'findings' and 'data' in an online list of key definitions that we have placed on the CERQual website. This is flagged in this paper, and the other papers in this series, under 'Availability of materials'. In addition, we will highlight this list of definitions in paper 1 of the series. We view data from qualitative primary studies as including all interpretations reported in a paper or report. This would include the primary study authors' analysis and interpretations and direct quotations from people who participated in the research. We have tried to make this clear in the definitions mentioned above.
* Very helpful clarification of what you mean by 'outlier' data and 'disconfirming cases'.	Thank you
'Balancing the coherence and utility of review findings'	
* Two examples are provided of ways authors might strengthen the coherence and limit the utility of their review findings: 1) reframing findings vaguely/narrowly, and 2) favouring descriptive over interpretive/explanatory findings. These are both explained nicely.	Thank you
* It would be helpful if, in the paragraph immediately following with a bulleted list of situations in which coherence might be found to be low, that the word 'utility' is explicitly used in the explanation of why these are circumstances 'important to review'. The consistent use of this terminology helps guide readers, particularly those who might be new to the topic.	We have made two additions to the sentences in that paragraph to reinforce the concept of utility.
'Guidance on how to assess coherence in the context of a review finding': * The text is clear in that the steps for assessing coherence might be iterative, however this is not reflected in the accompanying visual (Figure 3). The figure should be revised to reflect the potential for an iterative process rather than reflecting what appears to be a simple, linear process. Perhaps the addition of dashed arrows looping back to previous steps or something similarly minimalistic would be appropriate to accommodate the style of the current figure.	We have edited Figure 3 to indicate that the steps for assessing coherence might be iterative, and this is also noted in the text.

* Step 1: particularly given the direction of this paragraph describing accessing 'underlying data', the earlier comment addressing a clearer explanation of what is meant by the 'data' is needed at the beginning of the paper.	See above
* Very nice explanation of the three types of theories potentially present in qualitative evidence syntheses.	Thank you
* Strong section on three types of threats to coherence - the concrete example about stigma and hiding mental health status serves as a really useful tool to illustrate for readers a situation when gaps are not as concerning when researchers import a theory.	Thank you
* Table 1 and its examples/explanations are very useful and an excellent addition to this paper.	Thank you

Resubmission

29 Jun 2017 Submitted Manuscript version 2

9 Oct 2017 Author responded Author comments - Heather Munthe-Kaas

General comments from the series editor	Author responses and changes made
Thanks for providing more methodological detail in the overview and subsequent papers. There are still some areas where it would be better if you could provide further details to reflect the amount of international developmental work undertaken e.g. databases searched, timeframes, how literature reviewed etc.	We have added further detail to the overall methods description in paper 1 of the series. Specifically, we have:  - Included the years during which we ran workshops and seminars to obtain feedback on CERQual, and the numbers of workshops and presentations undertaken - Specified the period during which small group feedback sessions were run - Specified the number of CERQual users and Project Group members interviewed In the component papers (papers 3-6), we have noted that the literature searches that we undertook were informal in nature, as follows (example from paper 5): "When developing CERQual's adequacy component, we undertook informal searches of the literature, including Google and Google Scholar, for definitions and discussion papers related to the concept of adequacy and to related concepts such as data quantity, sample size and data saturation."

	We have also elaborated on the methods used to develop the content of paper 7 – please see below.
Ethics statements. Papers state that no humans were involved. Suggest amending to reflect consensus approach, interviews and questionnaires undertaken.	As we did not undertake formal data collection with people – all data collection was informal, in the context of training workshops, presentations and assessments of use of the approach, we have changed the ethics approval and consent to participate statements to the following: "Not applicable. This study did not undertake any formal data collection involving humans or animals."
Titles and papers could reflect paper nth of # part in a series.	We have changed all titles to the following format, as agreed earlier (example from paper 1): 'Applying GRADE-CERQual to qualitative evidence synthesis findings – paper 1 of 7: Introduction to the series'
State of the art has been removed from paper 6 but not all of the other papers in the series.	'State of the art' has been removed from all papers in the series.
The new figure outlining the process is a good addition. As a reader I would have found it easier to read papers 3-6/7 before reading paper 2.	As discussed by email with Liz Glidewell, we had a very long debate within the group about this and concluded that there is no perfect order because paper 2 (overall assessment) and papers 3-6 (components) need to be seen together. We placed 'overall assessment' before the component papers as we felt that readers needed to understand what they were working towards before understanding each component. We feel that it would be best to keep the order as it is, but have made the following changes to assist readers: Papers 2, 3, 4, 5 and 6: We have inserted text along the lines of the following (example from paper 2 (p6): 'These component papers are closely related to this paper on making an overall CERQual assessment of confidence and creating a Summary of Qualitative Findings table. We have placed this paper before the four CERQual component papers as we think that it will be helpful for readers to understand how the component assessments will be used before discussing the details of how to apply each component.' Papers 2, 3, 4, 5 and 6: We have included in each paper an additional table that brings together all of the key definitions from each of the papers.
Do you still want to publish paper 7 as a standalone or incorporate it into the overview along with the other ongoing research?	Yes, we feel that it works best as a standalone paper.

Would the figure in the introduction outlining the process work better across all papers in the series as it contains more information than the figure just outlining the 4 and probable 5 th component?	Thanks for this very helpful suggestion which we have implemented across all of the papers.
1. Introduction	
The lack of such methods constrains the use of...suggest reframing to "methods may constrain".	Change made
"The CERQual approach is intended to be applied to well conducted syntheses." Could this be confusing to those applying the four components? Isn't CERQual designed to provide evidence of confidence in a well conducted syntheses?	We have not found this to be confusing in our interactions with users of CERQual. We feel that there would be little point in applying CERQual to a synthesis that has been poorly conducted as the findings of such a synthesis are unlikely to be reliable and the synthesis is unlikely report transparently the methods used or to include sufficient information on the primary studies to allow a CERQual assessment to be undertaken. We take the same approach in relation to GRADE for effectiveness, for the same reasons. The problem is sometimes colloquially called 'garbage in-garbage out'!
The section "Applying CERQual across types of qualitative data and syntheses methods". Would this be better placed after outlining how CERQual was developed?	We agree and have moved this section.
"supported other teams". Can you say any more about the scale or settings involved?	We have provided more detail as follows: "Thirdly, we applied the CERQual approach within diverse qualitative evidence syntheses in the areas of health and social care [6-8, 26-33] and also supported other teams in using CERQual by providing guidance through face-to-face or virtual training meetings and commenting on draft Summaries of Qualitative Findings tables. At least ten syntheses were supported in this way (for example, [34, 35])."
Can you provided further detail about the questionnaire and qualitative interviews?	We have now provided further detail in the text and added an additional file listing the questions covered. The revised text reads as follows: "We then gathered structured feedback from early users of CERQual through an online feedback form that was made available to all CERQual users and through short individual discussions with six members of review teams and two members of the CERQual Project Group. The questions included in the online feedback form and individual discussions are available in Additional File X."
Summarise important areas for methodological research from table 4 in text for the readers ease?	We have revised the text as follows: "Table 4 identifies several important areas for further methodological research, including how to apply CERQual in syntheses that include qualitative and

	quantitative data; how best to present CERQual assessments together with other kinds of evidence; ways of applying CERQual to syntheses of sources that have not used formal qualitative research procedures; and whether CERQual requires adaptation for application to more interpretive synthesis outputs, such as logic models.”
2. Making an overall assessment and summary of qualitative findings	
Should the paragraph describing the four levels and rating down on p12 be moved to p10 under the 4 bulleted levels of concern?	This change has been made.
Place the text relating to variation in assessors after the text outlining who should undertake an assessment?	This change has been made.
Table 5. typo in component t missing.	This typo has been corrected.
Should you advise assessors to report how they've handled variation in levels of concern?	
3. Methodological limitations – problems design or conduct of primary studies	
Consider adding a brief description of the Evidence Profile to p12.	Ok. We have now added the following parentheses describing the evidence profile on page 12 following the sentence: “Where you have concerns about methodological limitations, describe these concerns in the CERQual Evidence Profile in sufficient detail to allow users of the review findings to understand the reasons for the assessments made (The Evidence Profile presents each review finding along with an explanation of its CERQual assessment)”
Link in text to table 2?	We have now added the following on page 9: “See Table 2 for an outline of areas where further work is needed with respect to critical appraisal tools for qualitative research.”
4. Coherence – How well finding supported by body of evidence 3500 3429	

Consider adding a brief description of the Evidence Profile to p13.	We have added a brief description of the evidence profile on page 12: “Where you have concerns about coherence, you should describe these concerns in the CERQual Evidence Profile in sufficient detail to allow users of the review findings to understand the reasons for the assessments made. The Evidence Profile presents each review finding along with the assessments for each CERQual component, the overall CERQual assessment for that finding and an explanation of this overall assessment. For more information, see the second paper in this series [19].”
5. Adequacy of data – degree of richness and quantity of data 3500 2507	
Consider contacting authors for further information as in other assessments?	We have added the following information to lines 204-205: “An overview of the number of studies from which this data originated, and where possible, the number of participants or observations. Information about the number of participants or observations supporting each finding may be difficult to gain from the individual studies. While most studies describe the number of participants they included in their study overall or give some indication of the extent of their observations, they may be less clear about how well represented participants are in different themes and categories. You can contact study authors for additional information, but they may not be able to readily provide this level of detail. In these cases, this lack of information should be noted, and your assessment of data adequacy will have to be made based on the information available.”
The sentence “For a description on descriptive and explanatory findings...” isn’t embedded.	We have moved this sentence to lines 232-233.
Consider adding a brief description of the Evidence Profile to p12.	We have added the following information to lines 277-279: The Evidence Profile presents each review finding along with the assessments for each CERQual

	component, the overall CERQual assessment for that finding and an explanation of this overall assessment.
6. Relevance – extent applicable to context (perspective or population, phenomenon of interest, setting) of review question 3500 3551	
I found a lot of the text more relevant to conducting a review than the CERQual assessment e.g. using theories and frameworks, how and when the review question should be developed, the pre-specification of sub-groups, strategies for identifying and selecting studies, trade-offs in searching.	Relevance is the only CERQual component that links directly to the review question. All the issues raised by the Editor need to be taken into consideration at the review design stage. We make this clear in the manuscript. See P6: 'Relevance is the CERQual component that is anchored to the context specified in the review question. How the review question and objectives are expressed, how a priori subgroup analyses are specified, and how theoretical considerations inform the review design are therefore critical to making an assessment of relevance when applying CERQual.' See page 11: 'When assessing relevance, you should reflect on how the sample was located and on the underpinning principles that determined its selection....'
Word missing p13 "You should if possible, that this"	Sincere apologies, this typo was corrected previously but the corrected draft was not uploaded last time.
Is it possible to comment on how the levels of concern map onto the different threats to relevance 'partial', 'indirect' and 'unclear'?	Tables 3, 4, 5 and 6 provide visual examples. Sincere apologies, these tables may not have been uploaded in error last time.
7. Dissemination bias – selective dissemination of studies or findings 2000 2455	
Methodological details e.g. 'consulting relevant literature' and 'additional empirical work'	We have added further detail as follows: Abstract: "We developed this paper by gathering feedback from relevant research communities, searching MEDLINE

		and Web of Science to identify and characterize the existing literature discussing or assessing dissemination bias in qualitative research and its wider implications, developing consensus through project group meetings, and conducting an online survey of on the extent, awareness and perceptions of dissemination bias in qualitative research.” Main text: “We used a pragmatic approach to develop our ideas on dissemination bias by consulting the literature on this topic, including searching MEDLINE and Web of Science to identify and characterize the existing literature discussing or assessing dissemination bias in qualitative research and its wider implications [3]; talking to experts in dissemination bias and qualitative evidence synthesis in a number of workshops; and developing consensus through multiple face-to-face CERQual Project Group meetings and teleconferences. We also undertook an online survey of researchers, journal editors and peer reviewers within the qualitative research domain on the extent, awareness and perceptions of dissemination bias in qualitative research [4].”
--	--	--

Resubmission 2

9 Oct 2017 Submitted Manuscript version 3

Publishing

17 Oct 2017 Editorially accepted

How does Open Peer Review work?

Open peer review is a system where authors know who the reviewers are, and the reviewers know who the authors are. If the manuscript is accepted, the named reviewer reports are published alongside the article. Pre-publication versions of the article and author comments to reviewers are available by contacting info@biomedcentral.com. All previous versions of the manuscript and all author responses to the reviewers are also available.

You can find further information about the peer review system here.